# NEAREST NEIGHBOR MACHINE TRANSLATION

**Urvashi Khandelwal**[†,*]**, Angela Fan**[‡]**, Dan Jurafsky**[†]**, Luke Zettlemoyer**[‡]**, Mike Lewis**[‡]
[†]Stanford University
[‡]Facebook AI Research
{urvashik,jurafsky}@stanford.edu
{angelafan,lsz,mikelewis}@fb.com

## ABSTRACT

We introduce $k$-nearest-neighbor machine translation ($k$NN-MT), which predicts tokens with a nearest neighbor classifier over a large datastore of cached examples, using representations from a neural translation model for similarity search. This approach requires no additional training and scales to give the decoder direct access to billions of examples at test time, resulting in a highly expressive model that consistently improves performance across many settings. Simply adding nearest neighbor search improves a state-of-the-art German-English translation model by 1.5 BLEU. $k$NN-MT allows a single model to be adapted to diverse domains by using a domain-specific datastore, improving results by an average of 9.2 BLEU over zero-shot transfer, and achieving new state-of-the-art results—without training on these domains. A massively multilingual model can also be specialized for particular language pairs, with improvements of 3 BLEU for translating from English into German and Chinese. Qualitatively, $k$NN-MT is easily interpretable; it combines source and target context to retrieve highly relevant examples.

## 1 INTRODUCTION

Non-parametric methods have recently been successfully applied to tasks such as language modeling (Khandelwal et al., 2020) and question answering (Guu et al., 2020; Lewis et al., 2020). They allow models that are (1) *expressive*, because they can use an arbitrary amount of data at test time; (2) *adaptable*, because predictions can be controlled by changing the datastore, and (3) *interpretable*, because the data used to make the prediction can be directly inspected. We introduce $k$NN-MT, a simple non-parametric method for machine translation (MT) using nearest neighbor retrieval. $k$NN-MT can be added to any pre-trained neural translation model without further training, and significantly improves performance for in-domain, out-of-domain, and multi-lingual evaluations.

More specifically, $k$NN-MT interpolates the target-token softmax distribution from a neural MT model with a multinomial generated using nearest neighbor search over examples cached in a data store. The cache is over translation contexts (i.e. the complete source and prefix of the target), and is indexed by hidden states computed from the base MT model. We hypothesize that contexts which are close in representation space are more likely to be followed by the same target word. We show this is not only true for the original training data, thereby improving base model performance, but across a range of different bi-text corpora, allowing for simple and effective model adaptation.

Our work builds upon recent results showing the effectiveness of nearest neighbor methods in unconditional language models (Khandelwal et al., 2020). We generalize to conditional language models, by using both source and target context, and show nearest neighbour models can be effective for generation in addition to density estimation. Compared to prior work on non-parametric methods for MT, our approach is arguably simpler (in that it requires no training, as compared to Gu et al. (2018)) and more expressive (in that it provides access to billions of key-value pairs during inference, as compared to Zhang et al. (2018); Gu et al. (2018)).

Extensive experiments show that $k$NN-MT scales to datastores containing billions of tokens, improving results across a range of settings. For example, it improves a state-of-the-art German-English translation model by 1.5 BLEU. $k$NN-MT can also be used to adapt a single model to

---

*Work done while the first author was interning at Facebook AI Research.

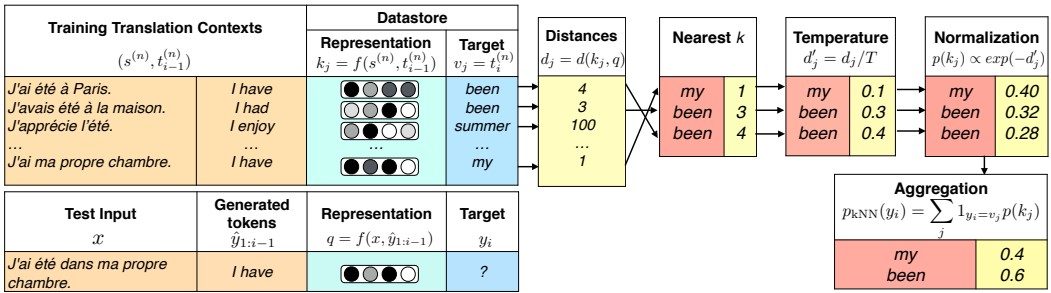

Figure 1: An illustration of how the $k$NN distribution is computed. The datastore, which is constructed offline, consists of representations of training set translation contexts and corresponding target tokens for every example in the parallel data. During generation, the query representation, conditioned on the test input as well as previously generated tokens, is used to retrieve the $k$ nearest neighbors from the datastore, along with the corresponding target tokens. The distance from the query is used to compute a distribution over the retrieved targets after applying a softmax temperature. This distribution is the final $k$NN distribution.

diverse domains by simply adding a domain-specific datastore—improving results by an average of 9.2 BLEU over the base model out-of-domain, and even outperforming existing models that train on these domains. Finally, language-pair-specific datastores are used to adapt a multilingual model to particular language pairs, with improvements of 3 BLEU for translating English into German and Chinese. We find that retrievals from $k$NN-MT are typically highly contextually relevant.

## 2    NEAREST NEIGHBOR MACHINE TRANSLATION

$k$NN-MT involves augmenting the decoder of a pre-trained machine translation model with a nearest neighbor retrieval mechanism, allowing the model direct access to a datastore of cached examples. The translation is generated word-by-word; at each time step, we find the most similar contexts in the datastore, and compute a distribution over the corresponding target tokens, as shown in Figure 1. This distribution is then interpolated with the output distribution from the pre-trained MT model.

More specifically, given an input sequence of tokens in a source language $s = (s_1, \ldots, s_{M_1})$, a neural MT model outputs a sequence of tokens $t = (t_1, \ldots, t_{M_2})$ in the target language. When using autoregressive decoders, the output distribution for each token $t_i$ in the target sequence is conditioned on the entire source sequence as well as the previous target tokens, $p(t_i | s, t_{1:i-1})$. Let $(s, t_{1:i-1})$ be the *translation context* and $t_i$ be the *target token*.

**Datastore creation**    Our datastore is constructed offline and consists of a set of key-value pairs. The key is a high-dimensional representation of the entire translation context computed by the MT decoder, $f(s, t_{1:i-1})$, where $f$ represents a mapping from input to an intermediate representation of the decoder. The value is the corresponding ground truth target token $t_i$. For a parallel text collection $(\mathcal{S}, \mathcal{T})$, the representations are generated by a single forward pass over each example and the complete datastore is defined as follows:

$$(\mathcal{K}, \mathcal{V}) = \{(f(s, t_{1:i-1}), \ t_i), \ \forall t_i \in t \mid (s, t) \in (\mathcal{S}, \mathcal{T})\} \tag{1}$$

Tokens from the source language are not stored directly as values in the datastore. Conditioning on the source is implicit via the keys, and the values are only target language tokens.

**Generation**    At test time, given a source $x$, the model outputs a distribution over the vocabulary $p_{MT}(y_i | x, \hat{y}_{1:i-1})$ for the target $y_i$ at every step of generation, where $\hat{y}$ represents the generated tokens. The model also outputs the representation $f(x, \hat{y}_{1:i-1})$, which is used to query the datastore for the $k$ nearest neighbors $\mathcal{N}$ according to squared-$L^2$ distance, $d$. In practice, the search over billions of key-value pairs is carried out using FAISS (Johnson et al., 2017), a library for fast nearest neighbor search in high-dimensional spaces.

The retrieved set is converted into a distribution over the vocabulary by applying a softmax with temperature $T$ to the negative distances and aggregating over multiple occurrences of the same vo-

cabulary item. Using a temperature greater than one flattens the distribution, and prevents overfitting to the most similar retrievals.

$$p_{\text{kNN}}(y_i|x, \hat{y}_{1:i-1}) \propto \sum_{(k_j, v_j) \in \mathcal{N}} \mathbb{1}_{y_i = v_j} \exp\left(\frac{-d(k_j, f(x, \hat{y}_{1:i-1}))}{T}\right) \tag{2}$$

While a pure $k$NN approach is effective, we improve results by interpolating with the base model distribution, which is more robust in cases without relevant cached examples. The model and $k$NN distributions are interpolated with a tuned parameter $\lambda$, resulting in the final $k$NN-MT distribution:

$$p(y_i|x, \hat{y}_{1:i-1}) = \lambda\, p_{\text{kNN}}(y_i|x, \hat{y}_{1:i-1}) + (1 - \lambda)\, p_{\text{MT}}(y_i|x, \hat{y}_{1:i-1}) \tag{3}$$

The complete translation is generated using beam search.

$k$**NN-MT vs. $k$NN-LM** $k$NN-MT is a generalization of $k$NN-LM applied to conditional sequence generation, with a few important differences. First, the keys are not only conditioned on prior context, but also on the source sequence (here, in a different language). This means that the representations must encode both source and target context; we show examples in Section 6. Second, there is an additional tuned parameter, the softmax temperature. Higher temperatures flatten the distribution and allow for greater diversity without overfitting to the retrieved contexts, as shown in Section 6.

## 3 EXPERIMENTAL SETUP

We experiment with $k$NN-MT in three settings: (1) single language-pair translation, (2) multilingual MT and (3) domain adaptation.

**Data**    We use the following datasets for training and evaluation.

WMT'19: For the single language-pair experiments, we use WMT'19 data for German-English.

CCMATRIX: We train our multilingual model on CCMatrix (Schwenk et al., 2019), containing parallel data for 79 languages and 1,546 language pairs. The parallel sentences are mined from cleaned monolingual commoncrawl data created using the ccNet pipeline (Wenzek et al., 2019). Semantically similar sentences in different languages are aligned using a learned distance measure; we use examples where the distance measure is at least 1.06, resulting in 4 billion sentence-pairs.

NEWSTEST: The newstest2018 and newstest2019 test sets from WMT (Bojar et al., 2018; Barrault et al., 2019) are used as validation and test sets for the multilingual experiments. The same German-English validation and test sets are also used for evaluation in the single language-pair and domain adaptation experiments.

TED TALKS: We use the Ted Talks data prepared by Qi et al. (2018) for evaluation in the multilingual setting, particularly to explore performance for language pairs that do no include English.

MULTI-DOMAINS: We use the multi-domains dataset (Koehn & Knowles, 2017), re-split by Aharoni & Goldberg (2020) for the domain adaptation experiments. It includes German-English parallel data for train/validation/test sets in five domains: Medical, Law, IT, Koran and Subtitles.

**Models**    For the single language-pair and domain adaptation experiments, we use the WMT'19 German-English news translation task winner (Ng et al., 2019), available via the FAIRSEQ library (Ott et al., 2019).[1] It is a Transformer encoder-decoder model (Vaswani et al., 2017) with 6 layers, 1,024 dimensional representations, 8,192 dimensional feedforward layers and 8 attention heads. Apart from WMT'19 training data, this model is trained on over 10 billion tokens of backtranslation data and fine-tuned on newstest test sets from years prior to 2018. In this work, we do not use ensembles or $n$-best reranking.

For multilingual MT, we trained a 418M parameter Transformer-based encoder-decoder model on the CCMatrix data for 100K updates. The model has embedding dimension 1,024, hidden dimension 4,096, 12 layers in both the encoder and decoder, with 16 attention heads. To balance the

---

[1]https://github.com/pytorch/fairseq/tree/master/examples/translation

training of different language pairs, which have various resource levels, we apply temperature up-sampling with $T = 5$ (Arivazhagan et al., 2019). The vocabulary is shared across all languages and consists of 128K subwords extracted using sentencepiece (Kudo & Richardson, 2018).[2] All results use case-sensitive detokenized BLEU, measured using SACREBLEU (Post, 2018).We provide the SACREBLEU signatures, along with details on the statistical power of our experiments, in Appendix C.

$k$**NN-MT**    In this work, we use a FAISS index to represent the datastore and search for nearest neighbors. The keys are stored in clusters to speed up search and quantized to 64-bytes for space efficiency (the full-precision keys are discarded). The index is constructed offline via a single forward pass over every example in the given parallel text collection. We use the 1024-dimensional representation input to the final layer feedforward network as the key. Building the index involves a training phase to learn the cluster centroids. We use 5M keys for learning 131K cluster centroids for the multilingual experiments, and 1M keys for 4K clusters for in-domain data in the domain adaptation experiments. During inference, we query the datastore for 64 neighbors while searching 32 clusters. The interpolation and softmax temperature parameters are tuned on the validation sets.[3]

**Computational Cost**    While $k$NN-MT does not add trainable model parameters, it does add some computational overhead. The primary cost of building the datastore is a single forward pass over all examples in the datastore, which is a fraction of the cost for training on the same examples for one epoch. During inference, retrieving 64 keys from a datastore containing billions of items results in a generation speed that is two orders of magnitude slower than the base MT system. Generation speed can be improved by searching fewer clusters, using smaller beams, or querying smaller datastores, with relatively minor trade-offs in performance, as we will see in Section 5. Developing faster nearest neighbor search tools remains an active area of research (Guo et al., 2020).

## 4 EXPERIMENTS

### 4.1 SINGLE LANGUAGE-PAIR TRANSLATION

To test whether $k$NN-MT can improve a model's ability to generalize from its training data, we first apply it to a state-of-the-art translation model, using a datastore containing only the original training set. We use a state-of-the-art German-English model as our base MT system, which scores 37.59 BLEU on the newstest2019 test set.[4] This is a highly competitive baseline – apart from the WMT'19 training data, the base model has also been trained on over 10 billion tokens of extra backtranslation data as well as fine-tuned on newstest test sets from previous years. Providing this heavily tuned base model with a datastore containing about 770M tokens of WMT'19 training data **improves performance by 1.5 BLEU to 39.08, without any additional training**. This result shows that even very strong translation models can be improved with test-time access to training sets.

### 4.2 MULTILINGUAL MACHINE TRANSLATION

Next, we apply $k$NN-MT to multilingual machine translation, to measure its ability to add capacity to a model when using very large training sets. For these experiments, we create datastores using subsets of the CCMatrix parallel data that the model has been trained on.

**Retrieving neighbors from same source language data**    Here, we build one datastore per language-pair being tested, using the training examples for that language-pair. Table 1 shows performance for the baseline and $k$NN-MT on 17 language-pairs from newstest2019. Retrieving neighbors results in up to 3 BLEU improvements for English-German, English-Chinese and Chinese-English, with an average improvement of 1.4 BLEU across all 17 pairs, without any additional training.

Table 1 also shows the sizes of each of the datastores. Datastore size and the increase in BLEU are only weakly correlated across languages, though within a language, a larger datastore is decidedly

---

[2]Evaluating our model on the recently released OPUS100 (Zhang et al., 2020) corpus improves upon the result in Zhang et al. (2020) by 0.4 BLEU, suggesting that it is a very strong baseline.

[3]Code for $k$NN-MT will be available at `https://github.com/urvashik/knnlm`.

[4]The winning system (scoring 40.8 BLEU) extends this model with ensembling and $n$-best reranking.

| | de-en | ru-en | zh-en | ja-en | fi-en | lt-en | de-fr | de-cs | en-cs |
|---|---|---|---|---|---|---|---|---|---|
| Test set sizes | 2,000 | 2,000 | 2,000 | 993 | 1,996 | 1,000 | 1,701 | 1,997 | 2,000 |
| Base MT | 34.45 | 36.42 | 24.23 | 12.79 | 25.92 | 29.59 | 32.75 | 21.15 | 22.78 |
| +$k$NN-MT | **35.74** | **37.83** | **27.51** | 13.14 | 26.55 | 29.98 | **33.68** | 21.62 | **23.76** |
| Datastore Size | 5.56B | 3.80B | 1.19B | 360M | 318M | 168M | 4.21B | 696M | 533M |
| | en-de | en-ru | en-zh | en-ja | en-fi | en-lt | fr-de | cs-de | Avg. |
| Test set sizes | 1,997 | 1,997 | 1,997 | 1,000 | 1,997 | 998 | 1,701 | 1,997 | - |
| Base MT | 36.47 | 26.28 | 30.22 | 21.35 | 21.37 | 17.41 | 26.04 | 22.78 | 26.00 |
| +$k$NN-MT | **39.49** | **27.91** | **33.63** | **23.23** | 22.20 | 18.25 | **27.81** | 23.55 | **27.40** |
| Datastore Size | 6.50B | 4.23B | 1.13B | 433M | 375M | 204M | 3.98B | 689M | - |

Table 1: Multilingual machine translation with $k$NN-MT. All test sets are from newstest2019, except *ja-en/en-ja* which are from newstest2020. Adding $k$NN-MT increases BLEU scores in all cases, and by over 3 points for *en-de*, *zh-en* and *en-zh*. Bold scores indicate significant results based on statistically powered experiments (Card et al., 2020).

| | **Ted Talks** | | | | | **Newstest2019** | | | **Avg.** |
|---|---|---|---|---|---|---|---|---|---|
| | de-ja | ru-ja | uk-ja | de-ru | de-zh | fr-de | cs-de | de-cs | |
| Test set sizes | 4,442 | 5,090 | 3,560 | 4,288 | 4,349 | 1,701 | 1,997 | 1,997 | - |
| Base MT | 10.11 | 9.69 | 8.36 | 17.24 | 20.48 | 26.04 | 22.78 | 21.15 | 16.98 |
| +$k$NN-MT (en-∗) | 11.08 | 10.42 | 9.64 | 18.02 | 21.22 | 27.85 | 23.71 | 21.74 | 17.96 |
| Datastore Size | 433M | 433M | 433M | 4.23B | 1.13B | 6.50B | 6.50B | 533M | - |

Table 2: Adding datastores with English source-side data can improve translation from other languages by an average of 1 BLEU, suggesting that our representations generalize over different source langauges. The model's representations of the source generalize across languages and make cross-lingual retrieval effective.

better, as shown in Section 5. This suggests that underlying factors, such as the quality of the parallel data used to populate the datastore, may also factor into the size of the improvements from $k$NN-MT.

Finally, we also observe that improvements for languages translated into English, on average 1.23 BLEU, are lower than improvements for languages translated from English, on average 1.94 BLEU. As English is the most frequent language in the base model's training data, this suggests that kNN-MT is particularly useful for improving decoders in languages that may be underfit during training.

**Retrieving neighbors using English as the source language**  Here, we construct datastores from training examples where English is the source language, and the target language is the language being tested. This setting is useful for rarer language pairs with less bi-text, and is related to pivoting (Utiyama & Isahara, 2007; Cohn & Lapata, 2007). Table 2 shows that on five pairs from the Ted Talks data and three from newstest2019, we find that $k$NN-MT improves performance by 1 BLEU on average. This result shows that the model's representations of the source generalize well enough across languages to make cross-lingual retrieval effective. Further investigation is needed to study the extent to which multilingual representations from related and unrelated languages can improve translation performance via $k$NN-MT.

## 4.3  DOMAIN ADAPTATION

We also measure the effectiveness of $k$NN-MT for domain adaptation, in which we use a domain-specific datastore to adapt the model, without further training. For these experiments, we use the German-English translation model from Section 4.1 as our base MT system and provide domain-specific data in the datastores. We also explore the effects of retrieving neighbors from a large amount of out-of-domain data as well as from a single multi-domain datastore.

| | Newstest 2019 | Medical | Law | IT | Koran | Subtitles | Avg. |
|---|---|---|---|---|---|---|---|
| Test set sizes | 2,000 | 2,000 | 2,000 | 2,000 | 2,000 | 2,000 | - |
| Aharoni & Goldberg (2020): | | | | | | | |
|    one model per domain | - | **56.5** | 59.0 | 43.0 | 15.9 | 27.3 | 40.34 |
|    one model for all domains | - | 53.3 | 57.2 | 42.1 | 20.9 | 27.6 | 40.22 |
|    best data selection method | - | 54.8 | 58.8 | 43.5 | **21.8** | 27.4 | 41.26 |
| Base MT | 37.59 | 39.91 | 45.71 | 37.98 | 16.30 | 29.21 | 33.82 |
| +$k$NN-MT: | | | | | | | |
|    in-domain datastore | 39.08 | 54.35 | **61.78** | 45.82 | 19.45 | **31.73** | **42.63** |
|    WMT'19 datastore | 39.08 | 40.22 | 46.74 | 40.27 | 17.99 | 29.23 | 34.89 |
|    all-domains datastore | 38.88 | 54.54 | **61.11** | **48.63** | 19.22 | **31.70** | **43.04** |
| Datastore Size (in-domain) | 770M | 5.70M | 18.3M | 3.10M | 450K | 159M | - |

Table 3: Domain adaptation using $k$NN-MT. The base MT system is trained on WMT'19 data which is also treated as the in-domain data for newstest2019. $k$NN-MT improves the base model by an average of 9.2 BLEU, without training, to achieve the best reported results on this task.

**Domain-specific datastores**  Table 3 shows the base MT system's in-domain performance on newstest2019, as well as zero-shot transfer to five other domains. $k$NN-MT significantly outperforms the base MT system in all settings. For the multi-domains dataset, $k$NN-MT improves the base MT model performance by an average of 9.2 BLEU, with improvements as large as 16 BLEU on Law and 14.5 BLEU on Medical, all without any further training. We also provide scores from Aharoni & Goldberg (2020) for models trained on in-domain data, those trained on all domains jointly, and those trained using the best-performing data selection method proposed by the authors. We find that $k$NN-MT also outperforms the best reported average on the multi-domains dataset by 1.4 BLEU.

**Out-of-domain and multi-domain datastores**  Table 3 also shows performance for retrieving neighbors from 770M tokens of WMT'19 data that the model has been trained on. While the average BLEU for the multi-domain data is 1 point higher, the improvements are much smaller compared to using in-domain data. This illustrates the value of adding domain-specific data to the datastore over adding a large amount of arbitrary data. We also measure the effectiveness of building a single multi-domain datastore containing parallel data from all six settings. Performance on IT improves by 3 BLEU but scores for the other domains are mostly the same. This shows that $k$NN-MT is robust to the presence of out-of-domain examples since retrieving neighbors from a datastore where large amounts of data is out-of-domain does not hurt performance relative to using only in-domain data.

## 5 Tuning kNN-MT

We investigate how key hyperparameters affect multilingual $k$NN-MT on validation data. We provide validation set BLEU scores as well as hyperparameter choices for our experiments in Appendix A.

**Softmax temperature**  A softmax temperature is used when estimating the nearest neighbor distribution in order to prevent the model from assigning most of the probability mass to a single neighbor, thus hurting diversity. Values greater than 1 will flatten the distribution, which can improve $k$NN-MT performance. Figure 2 shows that a temperature of 1 results in significantly lower BLEU scores. For all of our experiments, values of either 10 or 100 prove to be optimal.

**Number of neighbors per query**  In our experiments, we fix the value of $k$, the number of neighbors retrieved per query, to 64. For a fixed temperature and interpolation parameter, we find that performance does not improve when retrieving a larger number of neighbors, and in some cases, performance deteriorates. This suggests that retrieving more neighbors can add noise to the sequence generation process. Figure 2 shows that in some cases, performance improves when retrieving fewer neighbors, and further gains may be possible by tuning this parameter.

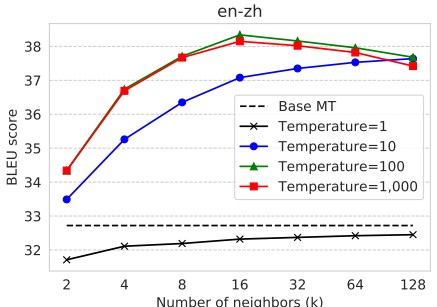
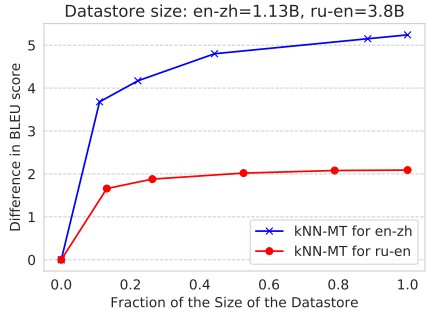

Figure 2: Effect of the number of neighbors retrieved and the softmax temperature on the validation BLEU score for *en-zh*. Temperatures greater than 1 are important to prevent the model from overfitting to the most similar neighbor. For higher temperatures, more neighbors do not always result in improvements.

Figure 3: Effect of datastore size on the validation BLEU score for *ru-en* and *en-zh*. Performance improves monotonically with size but retrieval can be slow for datastores containing billions of tokens. Smaller datastores, which account for a large fraction of the improvement, can be used for faster retrieval.

**Datastore size**  Figure 3 shows increasing the size of the datastore improves translation performance. However, larger datastores result in slower retrieval, indicating a speed-performance trade-off. Much of the benefit can be realized with much smaller, and correspondingly faster, datastores.

## 6  QUALITATIVE ANALYSIS

To better understand $k$NN-MT, we examine the retrievals for several examples. We use the German-English model and generate with only the $k$NN distribution ($\lambda = 1$) with beam size 1, retrieving $k = 8$ neighbors from the News Commentary and Common Crawl subsets of WMT'19 data.

Figure 4 shows an example from newstest2018 where all the retrieved neighbors map to the same target, *military*. Many of the retrieved examples include phrases similar to *tamed the military* such as *Autorität gegenüber dem Militär*, *Kontrolle des Militärs* and *das Militär gezwungen* on the source side and *authority over the military*, *control over the military* and *forced the military* given the local target side context, but differ sharply in their longer context, often describing different nations and centuries. We provide additional examples illustrating this point in Appendix B.

Another interesting observation is that $k$NN, even when not interpolated with the base model, is able to reconstruct named entities that are split into multiple subword tokens, even if that particular named entity does not appear in the datastore. One such example is the name *Haysom* that is split into subwords *Hay* and *som*. The retrieved neighbors for the first subword token include examples that contain the names *Hayes* and *Haydn*, while those for the second include *Grissom* and *Folsom*, showing subword representations are used effectively in the nearest neighbor search.

## 7  RELATED WORK

**Retrieval in Translation**  Recent work has integrated retrieval of words and phrases into neural translation, to gain some of the advantages of the previous generation of word- and phrase-based methods (Brown et al., 1993; Koehn et al., 2003). For example, Zhang et al. (2018) proposed guiding models by retrieving $n$-grams and up-weighting the probabilities of retrieved tokens. Tu et al. (2018) use cache-based models (Grave et al., 2017a;b) to save and retrieve translation histories, so models can adapt to changing contexts. Compared to these, $k$NN-MT has several advantages — for instance, the external datastore only needs to be created once, whereas the cache model requires constant writes. Further, $k$NN-MT scales retrieval to orders-of-magnitude larger datastores, while taking advantage of neural context representations.

Other work has retrieved complete example translation sentences at test time. Nagao (1984) proposed example-based MT for translating sequences by analogy. Before deep learning was widely

**Test Input**: *Dabei schien es, als habe Erdogan das Militär gezähmt.*
**Generated tokens**: *In doing so, it seems as if Erdogan has tamed the*

| Training Set Translation Context (source and target) | | Training Set Target | Context Probability |
|---|---|---|---|
| *Dem charismatischen Minis- terpräsidenten Recep Tayyip Erdoğan, der drei aufeinanderfol- gende Wahlen für sich entscheiden konnte, ist es gelungen seine Autorität gegenüber dem Militär geltend zu machen.* | *The charismatic prime minister, Re- cep Tayyip Erdoğan, having won three consecutive elections, has been able to exert his authority over the* | military | 0.132 |
| *Ein bemerkenswerter Fall war die Ermordung des gemäßigten Pre- mierministers Inukai Tsuyoshi im Jahre 1932, die das Ende jeder wirklichen zivilen Kontrolle des Militärs markiert.* | *One notable case was the assas- sination of moderate Prime Minis- ter Inukai Tsuyoshi in 1932, which marked the end of any real civilian control of the* | military | 0.130 |
| *Sie sind Teil eines Normal- isierungsprozesses und der Her- stellung der absoluten zivilen Kontrolle über das Militär und bestätigen das Prinzip, dass niemand über dem Gesetz steht.* | *They are part of a process of nor- malization, of the establishment of absolute civilian control of the* | military | 0.129 |
| *Diese hart formulierte Erklärung wurde als verschleierte, jedoch un- missverständliche Warnung ange- sehen, dass das Militär bereit wäre einzuschreiten...* | *That toughly worded statement was seen as a veiled but unmistakable warning that the* | military | 0.123 |
| ... | ... | ... | ... |

**Final *k*NN distribution**: military = 1.0
**Final Translation**: In doing so, Erdogan seemed to have tamed the military.
**Reference**: In doing so, it seems as if Erdogan has tamed the military.

Figure 4: Example retrievals using kNN-MT. Not only do the retrievals all correctly predict the target word *military*, but the local contexts tend to be semantically related. Both the source and the three nearest retrievals express the concept of control over the military.

adopted, this approach was extended to identifying portions of the training data that could be a trans- lation based on edit distance (Doi et al., 2005), matching training examples based on local trigram contexts (van den Bosch et al., 2007), using phrase-based memories (van Gompel et al., 2010) and incorporating syntactic features when retrieving similar examples (Stroppa et al., 2007; Haque et al., 2009). Recently, Gu et al. (2018) proposed a model that retrieves examples similar to the test source sequence and then attends over this subset of retrieved source-target pairs at the token level, while generating translations. Bulte & Tezcan (2019) and Xu et al. (2020) use fuzzy-matching with trans- lation memories and augment source sequences with retrieved source-target pairs. These techniques face challenges in identifying relevant retrieval candidates, as they focus on sentence-level retrieval. In contrast, $k$NN-MT focuses on token level retrieval from billions of key-value pairs, meaning that each word can retrieve the most relevant examples for its translation.

Finally, various studies have explored retrieving additional information to improve domain adap- tation, often using lexicons (Hu et al., 2019), domain-adaptive training (Farajian et al., 2017) or attending over neighbors similar to $n$-grams in the source (Bapna & Firat, 2019). These modifica- tions require additional training, whereas $k$NN-MT provides the flexibility to use different datastores when decoding in different domains, keeping the model fixed.

**Retrieval in Text Generation** Retrieval mechanisms have also been applied to generation tasks more broadly. Weston et al. (2018) and Fan et al. (2020) improve dialogue response generation sys- tems by retrieving examples and concatenating them to model inputs. Lewis et al. (2020) improve

open-domain question answering systems by retrieving relevant contexts from Wikipedia and concatenating them to the inputs. Hashimoto et al. (2018) use a retrieve-and-edit framework to generate structured outputs such as code, by jointly training the editor and retriever. For $k$NN-MT, retrieval results in a distribution over the vocabulary that is used for generation directly and does not require further training or providing the retrieval candidates as input.

## 8 CONCLUSION

We introduced a simple and effective method that can be applied to any neural MT model without further training. We show that similar contexts in a model's embedding space are more likely to be followed by similar next words, allowing the model to be improved by interpolation with a nearest neighbor classifier. The approach improves a state-of-the-art model in-domain, leads to large gains out-of-domain, and can specialize a multilingual model for specific language-pairs. Future work should improve efficiency, for example by down-sampling frequent target words in the datastore.

ACKNOWLEDGMENTS

The authors thank Kartikay Khandelwal for thoughtful discussions, Holger Schwenk and Sergey Edunov for sharing data and model details, and Matthijs Douze for answering questions about FAISS.

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

| | de-en | ru-en | zh-en | ja-en | fi-en | lt-en | de-fr | de-cs | en-cs |
|---|---|---|---|---|---|---|---|---|---|
| Test set sizes | 2,998 | 3,000 | 3,981 | 1,998 | 3,000 | 2,000 | 1,512 | 3,000 | 2,983 |
| Base MT | 38.21 | 30.51 | 21.93 | 14.44 | 21.50 | 28.68 | 28.46 | 21.97 | 20.66 |
| +*k*NN-MT | **40.48** | **32.6** | **24.49** | **15.62** | 22.11 | 29.41 | **29.74** | **23.01** | **21.79** |
| Datastore Size | 5.56B | 3.80B | 1.19B | 360M | 318M | 168M | 4.21B | 696M | 533M |
| Interpolation ($\lambda$) | 0.6 | 0.5 | 0.4 | 0.4 | 0.2 | 0.2 | 0.6 | 0.4 | 0.3 |
| Temperature ($T$) | 10 | 10 | 10 | 10 | 10 | 10 | 100 | 100 | 10 |

| | en-de | en-ru | en-zh | en-ja | en-fi | en-lt | fr-de | cs-de | Avg. |
|---|---|---|---|---|---|---|---|---|---|
| Test set sizes | 2,998 | 3,000 | 3,981 | 1,998 | 3,000 | 2,000 | 1,512 | 3,000 | - |
| Base MT | 39.07 | 26.00 | 32.72 | 16.31 | 16.02 | 21.11 | 25.16 | 24.16 | 25.11 |
| +*k*NN-MT | **42.22** | **29.52** | **37.96** | **18.28** | **17.22** | **22.84** | **26.39** | 24.5 | **26.95** |
| Datastore Size | 6.50B | 4.23B | 1.13B | 433M | 375M | 204M | 3.98B | 689M | - |
| Interpolation ($\lambda$) | 0.6 | 0.7 | 0.7 | 0.6 | 0.4 | 0.4 | 0.5 | 0.4 | - |
| Temperature ($T$) | 100 | 10 | 100 | 10 | 10 | 10 | 10 | 100 | - |

Table 4: Multilingual machine translation with *k*NN-MT on the validation set. We show the the tuned interpolation parameter ($\lambda$) as well as the tuned softmax temperature ($T$) for each language pair.

| | Newstest 2019 | Medical | Law | IT | Koran | Subtitles | Avg. |
|---|---|---|---|---|---|---|---|
| Test set sizes | 2,000 | 2,000 | 2,000 | 2,000 | 2,000 | 2,000 | - |
| Base MT | 48.07 | 39.94 | 45.78 | 35.78 | 16.30 | 29.74 | 33.51 |
| +*k*NN-MT: | | | | | | | |
|   in-domain datastore | 48.57 | **53.12** | **61.58** | **42.41** | **19.67** | **32.28** | **41.81** |
| Datastore Size (in-domain) | 770M | 5.70M | 18.3M | 3.10M | 450K | 159M | - |
| Interpolation ($\lambda$) | 0.4 | 0.8 | 0.8 | 0.7 | 0.8 | 0.7 | - |
| Temperature ($T$) | 100 | 10 | 10 | 10 | 100 | 10 | - |

Table 5: Domain adaptation using *k*NN-MT on the multi-domains validation data and newstest2018. The base MT system is trained on WMT'19 data which is also treated as the in-domain data for newstest2018. We present the interpolation ($\lambda$) and softmax temperature ($T$) hyperparameter choices for each domain.

## A  HYPERPARAMETER TUNING

In this section, we present validation set results as well as the hyperparameter choices for the multilingual machine translation and domain adaptation experiments. Only two hyperparameters have been tuned on the validation sets, the interpolation parameter $\lambda$ and the softmax temperature $T$. The number of neighbors $k$ has been fixed to 64, the number of clusters searched has been set to 32 and the beam size has been set to 5. For the number of clusters in the FAISS index, preliminary experiments showed that for larger datastores, while using more clusters does not hurt performance, it does significantly speed up the search process since searching within the clusters is exhaustive. Hence, we use 131K clusters for the multilingual experiments.

Table 4 shows the validation set BLEU scores for the multilingual experiments as well as the hyperparameter choices, and Table 5 shows the same for the domain adaptation experiments using only the in-domain datastores. Values for the interpolation parameter lie between 0 and 1. We also note that for a fixed value of $\lambda = 0.5$, using *k*NN-MT either performs similarly to or improves the base MT model's performance, but never hurts, on validation sets across the 17 language pairs evaluated in Section 4.2. For the temperature, we find that values of either 10 or 100 are optimal for all of our experiments.

**Test Input**: *Aber in Papua hat sich wenig verändert, und heute fühlen sich die Einheimischen betrogen.*
**Generated tokens**: *But not much has*

| Training Set Translation Context | Training Set Target | Context Probability | |
|---|---|---|---|
| *Nach einem schwer umkämpften Wahlkampf , der deutlich über zwei Milliarden Dollar kostete, sieht es für viele Beobachter aus, als hätte sich in der amerikanischen Politik nicht viel geändert...* | *After a hard-fought election campaign, costing well in excess of $2 billion, it seems to many observers that not much has* | changed | 0.143 |
| *Geändert freilich hat sich wenig: Schlecht gehandhabte Kriege...* | *But not much has* | changed | 0.137 |
| *Kaum etwas hat sich verändert , außer dass es jetzt nicht mehr die Bewohner des Appartement-Gebäudes...* | *Not much has* | changed | 0.130 |
| *Es ist zwar richtig, dass sich seit dem Ausbruch der globalen Finanzkrise vor über vier Jahren und der schon 2010 verabschiedeten Dodd-Frank-Finanzmarkreformen in den Vereinigten Staaten kaum etwas daran geändert hat...* | *True, while the global financial crisis erupted more than four years ago, and the Dodd-Frank financial reforms were adopted in the United States back in 2010, not much has* | changed | 0.121 |
| ... | ... | ... | |

**Final $k$NN distribution**: changed = 1.0
**Final Translation**: But not much has changed in Papua, and locals feel betrayed today.
**Reference**: But precious little has changed in Papua, and today local people feel betrayed.

Figure 5: An example where $k$NN-MT retrieves the same target token, *changed*, across all the neighbors. It matches local contexts on both the source and target sides, but not the global context regarding *Papua*.

## B  ADDITIONAL EXAMPLES

Figure 5 further illustrates the behavior of $k$NN-MT using local contexts in both the source and target to retrieve nearest neighbors. Figure 6 shows a case where the model has very little target-side prior context and mainly relies on the source context to retrieve the best neighbors.

## C  BLEU SCORES

In this paper, all results use case-sensitive detokenized BLEU, measured using SACREBLEU (Post, 2018), with the following signatures:
General: `BLEU+case.mixed+numrefs.1+smooth.exp+tok.13a+version.1.4.13`
For chinese: `BLEU+case.mixed+numrefs.1+smooth.exp+tok.zh+version.1.4.13`
For japanese:
`BLEU+case.mixed+numrefs.1+smooth.exp+tok.ja-mecab-0.996-IPA+version.1.4.13`

For Table 1 and other experiments, we follow advice from Card et al. (2020) regarding the statistical power of machine translation experiments given the improvements in BLEU scores and the size of the dataset. The authors present results for a single language pair and we verify that their assumptions hold for a couple of other language pairs. Specifically, we find that for Chinese-English $P_0 = 0.13$ and $b_0 = 12$, and for English-Chinese $P_0 = 0.07$ and $b_0 = 16$. This indicates that these experiments, with the test sets containing about 2,000 examples, have close to 100% power which was verified using the notebooks provided by Card et al. (2020). We refer the reader to the original paper for more details. More generally, experiments on datasets which contain about 2,000 examples, with improvements of about 1 BLEU or higher, are statistically powered.

**Test Input**: *Wir werden das Beste tun, mit dem, was wir haben.*
**Generated tokens**: *We*

| Training Set Translation Context (source and target) | | Training Set Target | Context Probability |
|---|---|---|---|
| *Wir werden versuchen zu beweisen, dass die Vermutung falsch ist, dies zu tun, nur um ein Gegenbeispiel Leinwände, dass die Aussage falsch ist, zu finden.* | *We* | will | 0.145 |
| *Allerdings, wenn man sich diese große Nation, wird diese Nation aussehen zu weit, und wir werden das tun, was wichtig und wertvoll, Identity Wiederherstellen der Nation.* | *However, if you look at this great nation, this nation will look too wide and we* | will | 0.132 |
| *Wir werden alles tun, um die Dinge für die Anfänger sehr einfach zu machen, während wir es den Experten erlauben, Dinge zu verändern, falls sie wollen.* | *We* | will | 0.127 |
| *"Wir werden ihre Fälle und die Fälle anderer politischer Gefangener vor die Gerichte bringen und das falsche Bild zerstören...* | *"We* | are | 0.127 |
| ... | ... | ... | ... |

**Final kNN distribution**: will = 0.639, are = 0.238, intend= 0.123
**Final Translation**: We will do the best we can with what we have.
**Reference**: We'll do the best we can with what we got.

Figure 6: An example where the model has access to a very short amount of target-side context that is ambiguous. kNN-MT is able to rely on source context to resolve this and generate the correct target token, *will*.

