# OpenReview forum: "Nearest Neighbor Machine Translation"
_ICLR.cc/2021/Conference — ICLR 2021 Poster_

### Official Review · AnonReviewer3 · 2020-10-27
**A new approach for NMT decoding that exploits a translation database of billions tokens with some improvements over state-of-the-art for a huge computational cost. Evaluation and analysis are very well designed and useful for future work.**

**Rating:** 6
**Confidence:** 5

**Review:**

This work presents an approach to exploits at decoding time a very large translation memory to improve NMT. An extensive evaluation is performed along with some detailed analysis on the important parameters.



Strengths:
- the idea is very simple, very easy to understand, and intuitive
- can be added to existing pre-trained NMT model
- many interesting applications (domain adaptation, multilingual model specialization for instance) are presented and are mainly the reason why I think paper can be accepted for publication.
- the paper is easy to read and well-written

Weaknesses:
-  exploiting a translation memory at test time is not novel (exploitation of billions of tokens is rather impressive but in my opinion making this possible is more an engineering problem)
- the approach is described within one page, the remainder of the paper is about evaluation and analysis. For ICLR, the paper lacks of substance.
- the improvements over SOTA English-German are very small considering that billions of tokens are exploited and the high decoding cost.
- the experiments presented in this paper are not reproducible since unpublishable data are exploited to train the system (eg. CCMatrix)
- computational cost at test-time is extremely high, as expected. This is probably why nobody tried it before. I do not see how it could be used for real-world applications. Focusing on reducing the computational cost would greatly improve the paper.


Questions/suggestions:
- "we also provide scores from Aharoni & Goldberg (2020)": did you check that these scores are comparable with yours? It is unclear in the paper whether they also used sacreBLEU (insert the sacreBLEU signature in a footnote in your paper to help future work reusing your scores)
- I recommend to add the decoding time in the tables and a description of the hardware used. Since the major issue of the proposed approach is its computational cost, adding the decoding time would probably encourage future work to try to improve it.

---

> ### Author Response · Authors · 2020-11-19
> **Response to Reviewer3**
>
> Hello Reviewer3,
>
> Thanks for your comments and suggestions.
>
> Please note the general comment to all reviewers regarding novelty and decoding speed.
>
> Translation memories have been used in the past, for instance Bulte & Tezcan, 2019; Xu et al. 2020 use fuzzy-matching, but we use memories directly as predictions in the NMT system by exploiting the contextual representations learned by the model, and without having to train parameters to use the memories.
>
> The +1.5 BLEU on the sota German-English model is, in fact, quite surprising given how strong and heavily tuned the base model is. It won the WMT19 translation task and uses a ton of extra backtranslated data (Section 4.1). We report +1.5 BLEU with no additional training and no extra data!
>
> CCMatrix is only used for Section 4.2, we asked the authors (Schwenk et  al., 2019) for that data. The rest of the experiments use data and models that are all available publicly.
>
> Aharoni and Goldberg did use SacreBLEU for their evaluations. We’ll include the following signatures in our next draft:
> General: BLEU+case.mixed+numrefs.1+smooth.exp+tok.13a+version.1.4.13
> For chinese: BLEU+case.mixed+numrefs.1+smooth.exp+tok.zh+version.1.4.13
> For japanese: BLEU+case.mixed+numrefs.1+smooth.exp+tok.ja-mecab-0.996-IPA+version.1.4.13
>
> We’ll also add some hardware details to the next draft!

---

### Official Review · AnonReviewer2 · 2020-10-28
**Extension of KNN-LM**

**Rating:** 4
**Confidence:** 5

**Review:**

Summary:

This submission introduces the kNN-MT approach for neural machine translation, which incorporates the memorize-catching spirit with the nearest neighbor classifier on a large datastore when generating the decoding sentences, together with the neural machine translation model for similarity search. No additional parameters are needed, but the inference cost increases. The authors conduct experiments are different settings, the single language pair translation, the multi-lingual machine translation, and the domain adaption translation. Results show that kNN-MT can easily improve translation performances by searching out related test sentences with non-trivial scores.

Comments:

Generally speaking, the submission is okay and the proposed approach has no big flaws, however, I feel hard to make this submission to be accepted. The main reasons or concerns are:
1. It is clear that this submission is a direct and straightforward extension of the previously published ICLR-2020 paper: kNN-LM. As the authors also clearly stated in the abstraction. Therefore, in terms of the contributions and differences, they are quite limited. The technique is almost the same, except the key is added with the source language sentence. The presentation of this paper is also similar to kNN-LM. The direct extension of the kNN approach from Language Model to Neural Machine Translation makes me feel hard to recommend, and this makes much more like a technique report of the method extension.
2. To say about the approach, I acknowledge that this method is effective, as the authors have done with multiple experiments. However, the computation cost is also high. The authors also discussed this in Section 3. It is hard for real-time systems to afford the increased inference cost as this approach made. The improved results with a little increased cost are okay, but too much is not a good choice. Though the authors mentioned there is a trade-off and I also acknowledge this, but it is still not clear what is a good trade-off.
3. Also, this method highly depends on the scale of the dataset, also the similarity between training and test dataset, if I understand correctly. This assumption can hold for high resource translations, but for low resource translation, this would be limited. This is another drawback of these search-based algorithms.

Minor question: What is the effect if $\lambda$ is varied?

Therefore, shortly speaking, I feel this paper is straightforward to extend from the previous paper (indeed this is the future work and the answer from the review comments of previous work). This concerns me a lot for another one in ICLR-2021.

---------------

Update:
I thank the authors to give responses to my points, especially the discussion about novelty. But I still feel the success of KNN for NMT is similar for LM, that's why a lot of works study on NMT are also work on LM. Since this KNN method only targets at the decoder side, same as LM model. Therefore, I still feel not novel enough.

---

> ### Author Response · Authors · 2020-11-19
> **Response to Reviewer2**
>
> Hello Reviewer2.
>
> Thanks for your comments.
>
> We address (1) and (2) in a general comment to all reviewers.
>
> For (3), regarding the point about low-resource translation, we didn’t explicitly explore this. Table 2 shows some results where datastores containing examples translated from English can help with translation in language pairs that contain much less data. It shows that cross-lingual retrieval is effective because the source representations generalize well across languages. We agree it would be interesting to test this on zero-shot translation pairs and low-resource pairs in the future!
>
> Regarding the scale of the dataset, Figure 3 shows a trend for this.
>
> Regarding the point about train/test similarity, note that any parallel data can be included in the datastore, not just training data. In the case of the domain adaptation experiments, we show a model that was trained on WMT data can be adapted to other domains simply by carefully selecting the data used in the datastore. Table 3 shows that caching WMT data for translation in the other domains is helpful (+1 Bleu average), but far less effective compared to using domain specific datastores (+9 Bleu average).
>
> The interpolation parameter is manually tuned and lies between 0 and 1. The selected values are reported in the appendix, alongside the dev scores. At 0, no kNN is used. At 1, only kNN is used which has surprisingly good performance, sometimes better than the base model, but generally worse than the interpolation.

---

> > ### Comment · AnonReviewer2 · 2020-11-19
> > **it is still an extension on another application**
> >
> > Thanks for the authors to provide the responses to my questions, especially for the novelty and the decoding speed. Let us just have more discussions.
> >
> > * Novelty: The authors mentioned a lot about different perspectives about the NMT scenarios, for example, multi-lingual NMT, domain-adaption NMT. Yes, these are important problems that NMT community is working on, but what most concerns me is the novelty of the  "method", the "algorithm", the "interesting points" inside this work, and the difficulties and differences that exist in this work. Compared with language modeling, NMT is a conditional language modeling in a broad view. Therefore, it still makes me the direct extension of KNN-LM to KNN-NMT is somehow trivial. If like this, there can be KNN-SpeechTranslation, KNN-TTS, KNN-Summarization, and so on. Besides, the authors present in the same way as KNN-LM. The importance is about the unique design of this paper, instead of different applications.
> >
> > * Decoding: I cannot argue much about the decoding speed as the authors replied "fast nearest neighbor retrieval is an open and active research problem". If this paper is like KNN-LM, which first presents such a search method to improve high-quality translation systems, then we can tolerate more about the decoding speed. However, since this is an application, people expect more improvements of this problem. In other words, if the authors indeed make difference to improve the low efficiency of KNN-NMT, then it would be better.
> >
> > * Though it remains not so clear to low-resource setting. For domain adaptation, how to ensure and control the cost of carefully selecting the domain specific data?

---

> > > ### Author Response · Authors · 2020-11-24
> > > **Discussion with R2 on novelty**
> > >
> > > Thanks for the engagement! It seems there is a legitimate disagreement over the relative importance of novelty in methods versus simplicity and strong empirical results. We argue that the success of nearest neighbor classifiers for language modeling does not make it obvious that this method would improve a heavily tuned state-of-the-art translation model trained on billions of tokens. In general, we feel that the simplicity of kNN is a key advantage and effectively applying this method to conditional generation, through carefully designed experiments, is a novel and valuable contribution. We hope that in the future the community will build on these results and will work towards making these methods more efficient.

---

### Official Review · AnonReviewer4 · 2020-10-29
**Novel method with a diverse set of strong results.**

**Rating:** 8
**Confidence:** 4

**Review:**

This paper describes a nearest-neighbor enhancement to NMT, where internal token-level context representations are used to index into a large data store to find relevant (source, target prefix) pairs. Since the index representation is taken from a pre-softmax representation in the decoder network, no additional training of the NMT model is required. The authors show a diverse range of strong results, from improvements using a data store over the model’s own training data, to improvement from using a collection of domain-specific corpora not present during training used for domain adaptation, to language specific collections to improve capacity of multilingual models. They are also able to show by example how the model makes MT more interpretable.

This is a very strong paper. It's well-written and easy to read, the method is very novel to MT, and the results are great. The method isn’t practical right now (decoding is two orders of magnitude slower), but it’s very interesting and thought-provoking. I can imagine it influencing a lot of work, even if the actual method doesn’t see a lot of use.

The only complaint that I could imagine raising against this paper is that the method is not particularly novel in light of recent work on nearest-neighbor language modeling, but in this day and age, with so many papers available, I think it’s actually very important to make these incremental stops in neighboring fields to make the connections explicitly clear. All the great experiments on multilingual MT and domain adaptation also help a lot. To their credit, the authors provide a concise section discussing the changes that needed to be made for the conversion to conditional language modeling (MT).

Small concerns:

The exp(d) in Figure 1 is missing a negative: exp(-d).

Table 1: what does the bolding indicate? It looks like statistical significance, but if so, please be clear about what test was used.

---

> ### Author Response · Authors · 2020-11-19
> **Response to Reviewer4**
>
> Hello Reviewer 4,
>
> Thanks for your comments! We’re so glad you liked the paper!!
>
> Thanks for catching that typo! We’ll fix it in the next draft.
>
> For Table 1 and other other experiments, we followed advice from [1] regarding statistical power of MT experiments given BLEU improvements and dataset size. They’ve presented results for a single language pair and we verified that their assumptions hold for a couple of other language pairs too! The reference seems to be missing in this draft - sorry about that. We’ll be sure to add it to the next draft.
>
> [1] Card et al. With Little Power Comes Great Responsibility. EMNLP 2020.

---

### Official Review · AnonReviewer1 · 2020-10-29
**The authors extend a method in LM to MT. The experiments show the method is effective in a range of settings.**

**Rating:** 4
**Confidence:** 3

**Review:**

The paper is an extension of [1]. The task in [1] is Language Modeling, while this paper is doing machine translation with the similar idea. The authors propose a non-parametric method for machine translation via a k-nearest-neighbor (KNN) classifier. Specifically, it predicts tokens with a KNN classifier over examples cached in a so-called datastore and this method can be applied to any pre-trained neural machine translation model without further training. The experiments show that it improves results across a range of settings (in-domain, out-of-domain, and multi-lingual evaluations).

Strengths:
+ The method is simple and can be applied to pre-trained neural machine translation model without further training.
+ The experimental results across a range of settings are effective.

Weaknesses:
- Although the method is simple and does not add trainable parameters, it add the computational cost. The authors mentioned the computational cost briefly but there are no detailed experiments. It would be good to see the authors add more analysis on the computational cost, for example, how it varies with k.
- Technical novelty over [1] seems to be incremental, where a large portion of the work is essentially regarding machine translation as a language modeling and applying the method in [1] to machine translation.


[1] Khandelwal, Urvashi, et al. "Generalization through memorization: Nearest neighbor language models." arXiv preprint arXiv:1911.00172 (2019).

---

> ### Author Response · Authors · 2020-11-19
> **Response to Reviewer1**
>
> Hello Reviewer1,
>
> Thanks for your comments. We’ve addressed points regarding novelty and computational cost in a general comment to all reviewers.

---

### Author Response · Authors · 2020-11-19
**Response to reviewers regarding novelty and decoding speed.**

Hello reviewers,

Thanks for your comments!

The reviews are much in agreement that we propose a simple and effective method for improving machine translation, but also that it is based on similar techniques applied to language modeling. In contrast to some submissions, the reviews also acknowledge that our presentation emphasizes similarities to previous work, does not add unnecessary complexity, and that it is up front about current limitations (particularly, decoding speed). While it may seem obvious in hindsight that this method would be so effective, it was not clear to us that such methods would work for sequence-to-sequence problems, and required the development of several crucial tricks to make it work so well.

We touch upon finer points, regarding novelty and decoding speed, below.

**Novelty:**
As Reviewer 4 has noted, the primary contribution of the work is in showing the promise of non-parametric methods, like nearest neighbor retrieval over an external memory, for neural machine translation. We do this through thoughtfully designed experiments that show how effective this approach can be for improving NMT systems across a number of axes.

(1) Domain adaptation is a challenging problem in NMT and we have shown how to make an NMT system effective in 5 new domains without any in-domain training, leading to a +9.2 BLEU average improvement over the base model’s zero-shot performance. This is, to the best of our knowledge, a first of its kind contribution.

(2) Improving multilingual NMT systems by adding language specific data to the memory, without any language specific fine-tuning or specialized training schemes, is a novel contribution.

(3) This method is able to improve a state-of-the-art German-English NMT model using only the training data provided, without any additional training, by 1.5 BLEU. This result is extremely surprising and could not have been predicted from previous results because the base model uses a ton of backtranslated data in addition to WMT’19 training data (Ng et al. 2019 in the paper).

(4) The fact that the retrieved examples can be easily inspected to explain the model's prediction, makes this method more interpretable than traditional NMT systems. We believe this is an important contribution to the field of machine translation. See Figure 4 in the paper.

That we could make this non-trivial effort seem effortless, like a “straightforward extension”, well we’ll take that compliment :)

We are grateful to Reviewer4 for noting the potential of this work to influence future research!

**Decoding speed:**
We discuss the computational costs of kNN-MT in Section 3, acknowledging that decoding is slower than current generation speeds (which have improved a great deal in recent years), but also acknowledging that fast nearest neighbor retrieval is an open and active research problem. Through our work, we make a strong case for the effectiveness of the method to convince the community it is worth investing the effort to make it efficient. Addressing the computational efficiency is not purely an engineering effort. Apart from making nearest neighbor search faster (Guo et al, 2020 in the paper), we could also make use of linguistic knowledge and inductive biases of the model to reduce the size of the datastore without losing performance.

We hope, as Reviewer4 posits, that this work will inspire others to explore avenues to make nearest neighbor retrieval for NMT more computationally efficient in the future.

---

### Decision · Program_Chairs · 2021-01-07
**Final Decision**

**Decision:**

Accept (Poster)

**Comment:**

This paper extends past work on kNN-augmentation for language modeling to the task of machine translation: a classic parametric NMT model is augmented with kNN retrieval from an external datastore. Decoder-internal token-level representations are used to index and retrieve relevant contexts (source + target prefix) that weigh-in during the final probability calculation for the next target word. Results are extremely positive across a range of MT setups including both in-domain evaluation and domain transfer. Reviews are thorough, but quite divergent. There is general agreement that the proposed approach is reasonable, well-motivated, and clearly described -- and further, that experimental results are both solid and relatively extensive. However, the strongest criticism concerns the paper's relationship with past work.  In terms of ML novelty, everyone agrees (including the paper itself) that the proposed methodology is a relatively simple extension of past work on non-conditional language modeling. However, two of the four reviewers strongly feel that, in light of the potentially prohibitive decoding costs, the positive experimental results are not sufficient to make this paper relevant to an ICLR audience given the lack of ML novelty. In contrast, another reviewer strongly takes an opposite stand-point:  rather, that the results will be extremely impactful to the MT subcommunity at ICLR since they are unexpected (i.e. that a non-parametric model might compete with highly-tuned NMT systems) and very positive across a range of domains and settings (i.e. in-domain, out-of-domain, multilingual) -- further, that the approach has substantial novelty in the context of MT where parametric models are the norm and that it might inspire substantial future work  (e.g. on efficient decoding techniques and further non-parametric techniques) given that it so drastically breaks the current MT mold. The final reviewer shares the concern of the former two about novelty, but is swayed by the experimental results and potential uses for the model (given kNN augmentation is possible without further training) and therefore votes for a marginal accept. After thorough, well-reasoned, and well-intentioned discussion between all four reviewers, the reviews land just barely in favor of acceptance, but with substantial divide. After considering the paper, reviews, rebuttal, and discussion I am swayed by the argument that (a) these experimental results are largely unexpected, (b) they are both extremely positive and offer a new trade-off between test and train compute in MT, and (c) that the paper may therefore inspire substantial discussion and follow-up work in the community. Thus I lean in favor of acceptance overall.